

# Counteractive effects of regional transport and emissions
# control on the formation of fine particles: a case study
# during the Hangzhou G20 Summit
Ying Ji[1], Xiaofei Qin[1], Bo Wang[1], Jian Xu[1], Jiandong Shen[3], Jianmin Chen[1], Kan Huang[1,2,4,*],
Congrui Deng[1,2,*], Renchang Yan[3], Kaier Xu[3], Tian Zhang[3]
[1]Shanghai Key Laboratory of Atmospheric Particle Pollution and Prevention (LAP3), Department of
Environmental Science and Engineering, Fudan University, Shanghai 200433, China
[2]Shanghai Institute of Eco-Chongming (SIEC), No.3663 Northern Zhongshan Road, Shanghai 200062,
China
[3]Hangzhou Environmental Monitoring Center, Hangzhou, Zhejiang 310007, China
[4]Institute of Atmospheric Sciences, Fudan University, Shanghai 200433, China
*Correspondence to*: K. Huang (huangkan@fudan.edu.cn); C. R. Deng (congruideng@fudan.edu.cn)
**Abstract.** To evaluate the effect of temporary emissions control measures on air quality during the 2016
G20 Summit held in Hangzhou, China, an intensive field campaign was conducted with focus on aerosol
chemistry and gaseous precursors from 15 August to 12 September, 2016. The concentrations of fine
particles were reduced during the intense emission control stages, of which the reduction of carbonaceous
matters was mostly responsible. This was mainly ascribed to the decreases of secondary organic aerosols
via the suppression of daytime peak SOC formation. Although the regional joint control was enacted
extending to the Yangtze River Delta region, the effect of long-range transport on the air quality of
Hangzhou was ubiquitous. Unexpectedly high NOx concentrations were observed during the control
stage when the strictest restriction on vehicles was implemented, owing to the contribution from
upstream populous regions such as Jiangsu and Shandong provinces. In addition, the continental outflow
via sea breeze triggered a short pollution episode on the first day of the G20 Summit, resulting in a
significant enhancement of the nitrogen/sulfur oxidation rates. After the Summit, all the air pollutants
evidently rebounded with the lifting of various control measures. Overall, the fraction of secondary
inorganic aerosols (SNA) in $PM_{2.5}$ increased as relative humidity increased, but not for the concentrations
of $PM_{2.5}$. Aerosol components that had distinctly different sources and formation mechanisms, e.g.
sulfate/nitrate and elemental carbon, showed strong correlations exclusively during the regional/long-





range transport episodes. The SNA/EC ratios, which was used as a proxy for assessing the extent of
secondary inorganic aerosol formation, were found significantly enhanced under transport conditions
from northern China. This study highlighted that the emission control strategies were beneficial for
curbing the particulate pollution whereas the regional/long-range transport may offset the local emission
control effects to some extent.
**1 Introduction**
Fine particulate matters (PM) are associated with air quality, public health, and the Earth's climate
(Garrett and Zhao, 2006;Liao et al., 2006;Menon et al., 2008;Tie and Cao, 2009;Kim et al., 2008). China,
especially its megacities, has experienced frequent and severe air pollutions during the past decade.
Severe air pollution episodes were often accompanied with high PM levels. The chemical compositions
of PM mainly consist of secondary inorganic aerosols (SIA) and organic matters (OM) that can be
differentiated into the primary organic aerosol (POA) and secondary organic aerosol (SOA). SIA
typically accounted for 40-50 % of the particulate masses during heavy pollution events and OM of 30-
40 % (Sun et al., 2016;Chen et al., 2015;Huang et al., 2012;Guo et al., 2014). During a historical regional
pollution episode, the mass ratio of SIA was over 1/3 at both megacities (Beijing and Shanghai) and the
remote region (Huaniao Isle over the East China Sea) (Wang et al., 2015a).
After the holding of the 2008 Olympic Games and 2014 APEC (Asia-Pacific Economic Cooperation)
Summit in Beijing, the G20 Summit (Group of Twenty Finance Ministers and Central Bank Governors)
in Hangzhou from 4 to 6 September 2016 was the biggest international event in China in recent years. It
is an international economic cooperation forum aiming at promoting open and constructive discussions
and research on substantive issues between developed and emerging market countries in order to seek
cooperation and promote international financial stability and economic sustainability. To improve the air
quality during the Summit, the government took strict control measures to reduce air pollutants emissions
from transportation, industry, construction sites, and power plants. Thus, it is an excellent opportunity to
conduct impact assessment of control measures on the atmospheric components. In addition, this
assessment is expected to deduce the sources of different air pollution components and provide references
for the prevention and control of air pollution in the future.



The components of air pollutants are affected by both emission sources and weather conditions (Wang and Dai, 2016;Liu et al., 2016a;Schleicher et al., 2012). During the 2008 Olympic Games and the 2014 APEC Summit, similar control measures were taken in Beijing and surrounding areas to achieve a good status of air quality. During the 2008 Beijing Olympics, the decrease of $PM_{2.5}$ mass was mainly due to the reduction of SIA, and the unexpected $PM_{2.5}$ increase during the emission control period may be related to poor weather conditions such as transport from the south and a small amount of precipitation (Li et al., 2013). In addition, the contribution of SIA increased while opposite for organics during the haze development, indicating that NOx emission control should be a priority for improving air quality in Chinese mega-cities (Pan et al., 2016). Particulate matters and street dust remained high through the Olympic period probably due to redistribution of existing sources, implying that the aim of zero pollution is not achievable in the short term (Qiao et al., 2016). Moreover, significant reductions of $NO_X$ and VOCs were observed in the first two weeks after the control measures were fully implemented. However, the levels of ozone, sulfate and nitrate in $PM_{2.5}$ increased and high levels of ozone may accelerate the oxidation of $SO_2$ to form sulfate (Wang et al., 2010). During the 2014 APEC Summit, all the aerosol components were significantly reduced while the $O_3$ concentration was still high (Wang et al., 2015b). Reductions of the precursors of secondary aerosols over regional scales were crucial and effective in mitigating PM pollution (Sun et al., 2016;Chen et al., 2015). From the perspective of remote sensing, the regional emission control strategy could significantly lower the extent of regional transport (Huang et al., 2015). When the local emission reduction was weakened, the variability of weather condition would play a more important role and the regional long-range transport became important (Xu et al., 2016). Therefore, the mechanism of the control measures and meteorological conditions on perturbing the levels of air pollutants are complex.

The formation mechanisms of severe pollution are complex and have not been clarified so far, as well as the formations of nitrate and sulfate. The formation of nitrate is generally dominated by three pathways. Under the ammonia-rich conditions, nitrate is formed mainly via the reactions of gaseous $HNO_3$ or nitric acid in droplets; while in ammonia-poor environments, heterogeneous hydrolysis of $N_2O_5$ on aerosol surfaces dominated (Schryer, 1982;Pathak et al., 2009;Russell et al., 1986;Richards, 1983) and this pathway primarily occurs at night with high RH and low temperature (Lin et al., 2006). In the daytime, NOx reacted with hydroxyl radicals to produce nitric and nitrous acids via complex photochemical reactions (Khoder, 2002). As for sulfate, the gas-phase oxidation of $SO_2$ by OH radical is the main



pathway of its formation, and the heterogeneous uptake of $SO_2$ on pre-existing particles or in cloud
droplets with oxidation by $H_2O_2$, $O_3$, $NO_2$ and metal ions is also important to form sulfate (Cheng et al.,
2016;He et al., 2014;Khoder, 2002;Sander and Seinfeld, 1976). Both gas-phase and heterogeneous
reactions are found responsible for the increase of fine particles, ultimately leading to the occurrence of
haze (Li et al., 2013;Pan et al., 2016;Wang et al., 2010). The secondary formation of SIA was found to
be related to heterogeneous aqueous reactions and was largely dependent on the ambient humidity (Wang
et al., 2012). Humidity plays a more important role in the rapid increase of nitrate than that of sulfate and
ammonium in $PM_{2.5}$ (Pan et al., 2016). The nitrate to sulfate ratios also exhibited dependence on relative
humidity (RH) and the daily variation of $PM_{2.5}$ tracked the pattern of RH in Beijing (Cheng et al., 2014).
In this study, meteorological parameters, gaseous precursors, and aerosol chemical components in
Hangzhou before, during, and after the G20 Summit were analyzed. Learning how the emission control
measures affected the chemical compositions, sources, and formation mechanisms of fine particles under
variable meteorological conditions during the control periods can systematically evaluate the
effectiveness of control measures and provide relevant basis for the improvement of environmental
quality. Although studies about the impact of emissions control on air quality have been widely studied,
there are some new findings in this study. Obvious increases of air pollutants even appeared during the
two most rigorous control stages, the regional and long-range transport had significant impact on the air
quality of Hangzhou. The formation mechanism of sulfate was different from nitrate with the dominance
of photochemical formation for sulfate but heterogeneous formation for nitrate. The implementation of
emission control measures had a significant impact on modifying the diurnal patterns of SOC.
**2 Methodology**
**2.1 Observational site**
The observational site (120.17° N, 30.29° E) in this study is on the roof (~ 20 m high) of a residential
building in Hangzhou, Zhejiang province. It is about 13 km from Hangzhou International Expo Center,
which is the main venue of the G20 Summit (Fig. 1). This site is surrounded by the residential buildings
in the north, south and east direction, and by several hospitals, with banks and convenience stores in the





west. It is representative of mixed emissions such as residential, traffic, etc. During the study period,
three zones with different emission control intensities were generally set up as shown in Fig. 1
**2.2 Instrumentation**
**2.2.1 Water-soluble ions**
Water-soluble components of airborne fine particles were continuously measured by an Ambient Ion
Monitor (URG-AIM9000D) during the entire study period. The system consists of the Steam Jet Aerosol
Collector (SJAC) and Ion Chromatography (ICS-2100, Dionex). Air flowed into the sampling tube at a
rate of 16.7 L min$^{-1}$. The sampling tube is equipped with a PM$_{2.5}$ cyclone cutting head, which can separate
out the particulate matters less than 2.5 microns in aerodynamic diameter. Part of the air passed through
a liquid diffusion denuder at the rate of 3 L min$^{-1}$ in order to remove the interfering gases (mainly SO$_2$
and HNO$_3$) and the rest of air was emptied. The air then mixed with the hot saturated water after entering
the steam generator and the mixing chamber, turning aerosol particles to grow into droplets. The enlarged
particles were separated by an inertial separator. After filtering, the aerosolized liquid was temporarily
stored in an aerosol sample collector. Until the analysis time, the collector automatically injected the
samples into the ion chromatograph. The aerosolized water-soluble ions collected on-line were measured
by two ion chromatographs through three-way device simultaneously.
The routine QA/QC included that all standard solutions were of excellent grade purity and re-prepared
monthly. The correlation coefficients (R$^2$) of the standard curve were greater than 99.9 %, excepting for
NH$_4^+$ of R$^2$>99.5 %. The flow rate of the AIM system was checked periodically and kept at 3 L min$^{-1}$.
**2.2.2 OC/EC**
Organic carbon (OC) and element carbon (EC) in PM$_{2.5}$ were measured using a Semi-Continuous OC/EC
analyzer (SUNSET Laboratory). Particles with an aerodynamic particle diameter less than 2.5 μm were
collected by the cyclone separator at a sampling flow rate of 8 L min$^{-1}$ and the sampling time was 40 min
per cycle. Air particles were collected on a circular quartz filter with a diameter of about 1.6 cm and an
effective sampling area of 2.0 cm$^2$. The volatile organic compounds (VOCs) were removed by a multi-
layer parallel organic denuder during sampling. After finishing the collection, we used the high purity





helium gas to purge pipeline of the system, then the NIOSH (National Institute for Occupational Safety
and Health) 5040 TOT (thermal-optical transmittance) was used for analysis within the duration of 15 ~
20 min. The carbonaceous matters collected on the quartz film were gradually pyrolyzed and catalytically
oxidized to $CO_2$ by the programmed temperature and thermo-optical method, and then quantified by a
non-dispersive infrared detector (NDIR). The temporal resolution of measurement was 1 h and the
OC/EC detect sensitivity (calculate as C) can reach $0.1\mu g$ $m^{-3}$. The instrument was calibrated with
methane standard gas for each monitoring cycle and the monthly standard sucrose solution was used to
calibrate methane standard gas.

**2.2.3 PM$_{2.5}$ and trace gases**

PM$_{2.5}$ was measured by a continuous particulate matter monitor (5030, Thermo, USA). A 43i SO$_2$ gas
analyzer and a 42i NO-NO$_2$-NOx analyzer were used to measure the concentrations of trace gases.

**2.3 Data Analysis**

**2.3.1. Air mass back trajectory**

The HYSPLIT (HYbrid Single-Particle Lagrangian Integrated Trajectory) model is a complete system
for calculating simple air mass backward trajectories to dispersion and complex deposition simulations
(Draxler and Rolph, 2012). To clarify the possible sources of various air pollutants, the hybrid single-
particle Lagrangian integrated trajectory HYSPLIT4 was run online at the NOAA ARL READY Website
(HYSPLIT4, 1997) using the meteorological data archives of Air Resource Laboratory (ARL). The
meteorological input data used in the model was obtained from NCEP's global data assimilation system
(GDAS). In this study, all back trajectories were calculated at 500m AGL (above ground level).

**2.3.2. Concentration Weighted Trajectory (CWT) analysis**

The concentration weighted trajectory (CWT) analysis (Hsu et al., 2003), a useful tool for source
identification, was performed to pinpoint the potential geographic source regions of air pollutants. It
should be noted that both air mass trajectory and CWT analysis are methods to reveal potential sources
regions. Compared to the air mass trajectory analysis, CWT has an additional advantage of presenting



the spatial distribution of potential sources regions. In this study, we combined air mass trajectory and
CWT to identify the source regions of specific air pollutants.
In the CWT method, each grid cell is assigned a weighted concentration by averaging the sample
concentrations that have associated trajectories crossing the grid cell as follows:
$C_{ij} = \frac{1}{\sum_{l=1}^{M} \tau_{ijl}} \sum_{l=1}^{M} C_l \, \tau_{ijl}$         (1)
where $C_{ij}$ is the average weighted concentration in the ijth cell, l is the index of the trajectory, M is the
total number of trajectories, $C_l$ is the concentration observed on the arrival of trajectory l, and $\tau_{ijl}$ is
the time spent in the ijth cell by trajectory l. A high value for $C_{ij}$ is implies that air parcels traveling over
the ijth cell would be, on average, associated with high concentrations at the receptor.
To eliminate the uncertainty of $C_{ij}$ is caused by low $n_{ij}$ values, every $C_{ij}$ is should be multiplied by
an arbitrary weight function $W_{ij}$ to get more accurate results. The weight function $W_{ij}$ to is defined as:
$$W(n_{ij}) = \begin{cases} 1.0 & 3n_{ave} < n_{ij} \\ 0.7 & 1.5n_{ave} < n_{ij} \le 3n_{ave} \\ 0.4 & n_{ave} < n_{ij} \le 1.5n_{ave} \\ 0.2 & n_{ij} \le n_{ave} \end{cases}$$
$n_{ave}$ represents the average number of trajectories in grid cells with trajectories passing through the
partition region; $n_{ij}$ is the number of all trajectories in the (i, j) cell.
**2.3.3. EC-tracer method**
Organic carbon (OC) and elemental carbon (EC) were the major components of fine particles ($PM_{2.5}$)
(Malm et al., 2004). EC was a product of the carbon-based fuel combustion process and was considered
entirely derived from primary emissions, while OC can be derived from both primary emissions and
second formation. SOC (secondary organic carbon) can be estimated using EC as a tracer as below
(Turpin and Huntzicker, 1994),
$POC = (OC/EC)_{pri} \times EC$         (2)
$SOC = OC - (OC/EC)_{pri} \times EC$         (3)
Where $(OC/EC)_{pri}$ is the OC/EC ratio of freshly combusted aerosols. To determine the values of
$(OC/EC)_{pri}$, it is first assumed that the $(OC/EC)_{pri}$ values varied continuously. Then we calculated



the corresponding SOC concentrations based on each hypothesized $(OC/EC)_{pri}$ and the correlation
coefficient ($R^2$) of the SOC and EC pair (i.e., $R^2$ (EC, SOC)). Thus, a series of $R^2$ (EC, SOC) values can
be plotted against the OC/EC ratios. Since the sources of EC and SOC were independent, the OC/EC
ratio corresponding to the minimum $R^2$ (EC, SOC) was considered to be $(OC/EC)_{pri}$ (Wu and Yu, 2016).
**3 Results and Discussion**
**3.1 Air quality and weather conditions during the whole study period**
The whole study period was divided into five stages: S1 (15-23 August), S2 (24-27 August), S3 (28
August-3 September), S4 (4-6 September), and S5 (7-12 September). S1 was the reference stage without
intense emissions control measures. S2 was the stage of industrial and construction emissions control. In
detail, the emission control on industries was implemented during 24-25 August. Enterprises in
Hangzhou were either temporally suspended or reducing productions. After 25 August, construction
activities were prohibited. S3 added restriction on the motor vehicles. The odd-even traffic rule was fully
implemented and vehicles from outside Hangzhou were prohibited entering the city. In addition,
transportation of dusty materials was not allowed during this period. S4 was the G20 Summit period,
which was the most stringent emission control stage. S5 was the post-G20 stage with all the control
measures lifted.

The time series of hourly $PM_{2.5}$, $PM_{10}$, and its precursors ($SO_2$, NOx) are illustrated in Fig. 2, together

with the meteorological parameters (i.e., wind speed (WS), wind direction (WD), temperature (T),
relative humidity (RH), radiation, and boundary layer height (BLH)). As shown in Fig. 2, the easterlies
dominated during S1, S4, and S5. The winds changed from easterlies to westerlies in S2 and turned to
be from the southwest in S3. The average wind speed from S1 to S5 was 1.34, 1.68, 1.30, 1.31, and 0.96
m s$^{-1}$, respectively. RH in S3 was obviously lower than the other stages while it reached high in S5.
Temperature were the highest in S1 and S2 then gradually decreased in S3, finally declining quickly in
S4 and S5. Radiation was high during the first three stages (mean value of 301, 357, and 433W m$^{-2}$),
especially in S3. It turned to be weaker during the last two stages (mean value of 204 and 215 W m$^{-2}$).
The variation of boundary layer height was as similar as radiation to some extent. It was high from S1 to
S3 but quickly became shallow in S4 and S5. S2 was the stage of industries and construction emission



control accompanied with the highest wind speed, the concentrations of NOx, $SO_2$, and PM all dropped
to low levels. It should be noted that although S3 added the emission control measures on motor vehicles,
the concentrations of NOx remained at relatively high levels even under favorable meteorological
conditions such as high wind speed, strong radiation, and low relative humidity. This phenomenon will
be discussed later. Since S4 was the most stringent emission control period, all the air pollutants were
greatly reduced although the meteorological conditions were unfavorable due to relatively low wind
speed and high RH. However, a short pollution episode occurred on the morning of September 4 with
the hourly $PM_{2.5}$ concentration exceeding 100 μg m$^{-3}$. After all the control measures were lifted in S5,
$PM_{2.5}$ rebounded associated with unfavorable weather conditions (i.e. low wind speed and BLH, weak
radiation, and high RH). The average concentrations of $PM_{2.5}$ during the five stages were 37.4, 31.8, 40.4,
35.0, and 49.5 μg m$^{-3}$, respectively. On the whole, the $PM_{2.5}$ concentrations during control stages were
lower than the reference and post-G20 stages.

**3.2 Diurnal profiles of $PM_{2.5}$ species and meteorological variables**

The diurnal variations of $PM_{2.5}$ major compositions, as well as the key meteorological parameters, were
demonstrated in Fig. 3 for all the five stages. As for the meteorological parameters, in general, RH, T,
and WS exhibited consistent diurnal trends among the five stages. RH was relatively low during daytime
and high during nighttime while temperature showed the opposite trend. Wind speed was relatively low
during the first half of the day and gradually increased in the afternoon.
During all the stages, NOx exhibited peak values at around 6:00~8:00 AM LT (Local Time) and
16:00~20:00 PM LT, corresponding to the morning and evening rush hours due to the enhanced vehicular
emissions. In S4 which was the G20 period, the evening peak of $NO_X$ was almost missing and this should
be attributed to the stringent emission control during that period. While in S5, in addition to the peaks
during the morning and evening rush hours, $NO_X$ showed significant enhancement around the daybreak
from around 21:00 PM to 3:00 AM LT. This was ascribed to the allowance of heavy-duty diesel trucks
into Hangzhou during night-time after G20. This phenomenon was also reflected by the corresponding
EC and OC peaks around the similar period. In contrast, the high concentrations of $SO_2$ and $SO_4^{2-}$ mainly
appeared around 6:00 AM to 18:00 PM LT, tracking well with the working hours. Power plants and
industries were the major contributors to $SO_2$ emissions and they were mainly operating during daytime.



An exception was noted that the diurnal variation of sulfate in S5 was different from the other four stages
and, its peak appeared in the early morning and night. The low sulfate levels during daytime were likely
due to the low secondary conversion rate associated with weak radiation and low temperature in this
stage. In addition, there were sustained precipitation events during daytime on 7 September and 9
September (Fig. S1), which could have reduced the sulfate concentrations during daytime to some extent.
The high levels of sulfate during daybreak and night may be related to the heterogeneous reaction due to
the high RH and PM. High PM concentrations in S5 provided enough surface area for the conversion of
sulfate under high RH conditions (Mattias Hallquist, 2016).
POC and SOC were differentiated in five stages based on the method described in Sect. 2.2.3. As
shown in Fig. 3, POC in all five stages maintained at certain levels without dramatic diurnal fluctuations.
In contrast, SOC in S1, S2, and S5 showed a tendency to increase starting from the early morning and
reached a maximum in the midday, indicating the photochemical formation of SOC. This is consistent
with previous studies that photochemical pathways were of importance for the formation of SOC (Wyche
et al., 2014;Liu et al., 2015;Kleeman et al., 2007;Xu et al., 2017). Unlike the three stages above, there
were noticeable absences of SOC peaks around the midday in S3 and S4, resulting in ambiguous diurnal
fluctuations. The stringent emission control measures should exert a significant impact on the SOC
formation due to the great reduction of its precursors. Furthermore, the concentrations of SOC showed a
positive relationship with temperature (Fig. S2). Under relatively low temperature, the concentrations of
SOC stayed at relatively low levels and increased greatly with the increase of temperature, indicating an
enhanced role of higher temperature in SOC formation in summer. In this regard, the relatively low
temperature in S3 and S4 may also explain the low SOC concentrations during these two stages. As for
the relationship between SOC and RH, no clear correlation was observed in this study, which was as
similar as that observed in Beijing (Zheng et al., 2015). Overall, we found that the emission controls had
an evident suppressing impact on the SOC formation and crucial meteorological parameters (e.g.
temperature and radiation) were also of importance.
**3.3 Aerosol chemical composition**
Fig. 4a shows the comparison of aerosol chemical components among the five stages. The major
components of $PM_{2.5}$ were identified as SNA ($SO_4^{2-}$, $NO_3^-$, and $NH_4^+$), EC, and OM, which together





accounted for approximate 60-80 % of the aerosol masses during different stages (Fig. 4b). The sum of
SNA, trace ions ($Na^+$, $K^+$, $Ca^{2+}$, $Mg^{2+}$, and $Cl^-$), EC, and OM decreased with different extents from S2 to
S4 compared to S1, demonstrating the effectiveness of emission control measures in Hangzhou and its
surroundings on the improvement of air quality. Of which, the decrease of OM was mostly responsible
with a reduction percentage of 32 %, 15 %, and 38 % from S2 to S4 compared to S1. The reductions of
EC were 21 %, 18 %, and 23 % from S2 to S4. This suggested that the emission control measures played
a significant role in reducing the carbonaceous aerosols. On the opposite, SNA increased 8 % and 43 %
in S3 and S4, respectively. This highlighted SNA was more enhanced during the emission control stages
under variable meteorological conditions. Specifically, the average concentrations of $SO_4^{2-}$ in S3 (5.4 μg
$m^{-3}$) and $NO_3^-$ in S4 (3.9 μg $m^{-3}$) were higher than those of S1 ($SO_4^{2-}$:4.4 μg $m^{-3}$; $NO_3^-$: 2.2 μg $m^{-3}$). Given
that both S3 and S4 were the intense emission control periods, the unexpected increases of secondary
aerosol components may be attributed to the long-range transport or unfavorable meteorological
conditions. More detailed analysis will be presented in Sect. 3.4. After the G20 Summit, the sum of SNA,
OM, and EC increased 42 %, 52 %, and 62 % compared to S2-S4, clearly demonstrating the negative
effect of lifting the emissions control measures on deteriorating the air quality.
Fig. 4a. further shows the mass ratios of $NO_X/SO_2$, $NO_3^-/EC$, and $SO_4^{2-}/EC$ at each stage. The ratio of
$NO_X/SO_2$ gradually decreased from S1 to S4 as the emission control measures were more intensified,
indicating that $NO_X$ emissions were more effectively abated relative to $SO_2$ emissions. The $NO_X/SO_2$
ratio rose to the highest in S5, owing to the lifting of emission control measures especially from the traffic
sector. The ratios of $NO_3^-/EC$ and $SO_4^{2-}/EC$ can be used to pinpoint the extent of secondary formation by
minimizing the effect of different meteorological conditions on the absolute concentrations of aerosol
components (Zheng et al., 2015). In other words, the ratios of $NO_3^-/EC$ and $SO_4^{2-}/EC$ can represent the
extent of the secondary reactions. As shown in Fig. 4a, the $SO_4^{2-}/EC$ ratios gradually increased during
the first three stages, followed by a slight decrease during S4 and S5. Generally, $SO_4^{2-}/EC$ ratios varied
within a narrow range of around 3-4, indicating the relatively stable reactions of $SO_2$ to $SO_4^{2-}$ in the five
stages. The $NO_3^-/EC$ variation showed a different pattern that it remained consistently low during the
first three stages and then showed a substantial increase during S4 and S5. $NO_3^-/EC$ ratios in S4 and S5
increased about 2-3 times than those from S1-S3. Moreover, the $NO_3^-/EC$ ratios were lower than $SO_4^{2-}$
/EC during the first four stages while it exceeded $SO_4^{2-}/EC$ in S5.



### 3.4 Process analysis in each stage

### 3.4.1. High aerosol species in S1

Fig. 5 shows the time-series of the major aerosol chemical components during the whole study period. In S1, most of the aerosol components maintained at high levels, especially for sulfate, EC, and OC. Since the weather conditions were characterized of well-developed BLH, high temperature, low RH, and moderate WS (Fig. 2), air pollutants were supposed to be subjected to efficient diffusion. However, relatively high concentrations of EC and OC, accompanied by the high concentration of $NO_X$ were observed, indicating strong emissions from the traffic sector in S1. In addition, concentrations of sulfate were also at high levels, suggesting the considerable impact from the power grid.

### 3.4.2. Substantial decreases of aerosol species in S2

S2 was the stage that implemented the industrial and construction emission control measures. Concentrations of SNA, EC and POC were significantly reduced, indicating the great benefits from the emission control strategy.

### 3.4.3. Influence from long-range transport in S3

A continuously increasing trend of particulate mass concentrations was observed in S3 (Fig. 2), including $SO_4^{2-}$, $NH_4^+$, EC, and POC (Fig. 5). The meteorological conditions in this stage were generally favorable for the diffusion of air pollutants as indicated by the low RH, strong radiation, and high BLH (Sect. 3.1). It could be visualized that the high pollution episodes tended to accompany high wind speed (Fig. S3), suggesting the increases of aerosol components may be attributable to the regional or long-range transport. The 72-h backward trajectory clustering analysis was performed during S3 (Fig. 6a). It is shown that most of the backward trajectories were related to the regional/long-range transport with a contribution of more than 60 %, while the rest of the backward trajectories were restrained within the local range. To further identify the potential source regions of specific air pollutants, we conducted the concentration weighted trajectory (CWT) analysis (Fig. 6b-6e). The results showed fairly consistent CWT spatial patterns for $NO_X$ and $NO_3^-$, i.e. high $NO_X$ and $NO_3^-$ hotspots were mainly derived from




Hebei province, Shandong province, Shanghai and the conjunction area of Anhui and Jiangsu provinces.
This could partly explain why the concentrations of $NO_X$ increased significantly in S3, which was the
stage that the motor vehicle emission control measures were fully implemented in Hangzhou. Compared
to $NO_X$ and $NO_3^-$, the potential source regions of $SO_2$ and $SO_4^-$ exhibited inconsistent spatial patterns. As
shown in Fig. 6d-6e, the $SO_2$ CWT plot indicated hotspots mainly from southern Hebei, Shandong, and
Jiangsu provinces, while the potential sources of sulfate were mainly ascribed to regions south of
Shanghai, i.e., Hunan and, Jiangxi provinces.
**3.4.4. Impact from continental outflow in S4**
S4 was the G20 Summit period, which was the most rigorous emission control stage. However, a high
particulate pollution episode occurred with the hourly $PM_{2.5}$ peak concentration of exceeding 100 μg m$^-$
$^3$ between 0:00-5:00 LT in the morning of 4 September, which was the first day of the G20 Summit.
Consistently, concentrations of the major aerosol components also increased substantially (Fig. 7a). If
this short pollution episode was absent, the average concentrations of $PM_{2.5}$, SNA, and OC during S4
could be lowered by 12 %, 12 %, and 3 %, respectively. Fig. 7c shows the 48-h air mass backward
trajectories during this pollution period and six backward trajectories were computed at 500 AGL from
22:00 LT on 3 September to 8:00 LT on 4 September (22:00, 0:00, 2:00, 4:00, 6:00, 8:00). It is shown
that the prevailing air masses were mainly from Shandong and, then passed over the East China Sea
before reaching Hangzhou. As shown in Fig.7a, Cl$^-$ had a dramatic increase from almost zero before 4
September to a peak value of 0.24 μg m$^{-3}$ in the morning of 4 September along with the increase of RH,
which further indicated the long-range transport route over the ocean. This implied that the
meteorological conditions should be favorable for the heterogeneous reaction pathway of secondary
aerosol formation facilitated by the humid sea breeze. As for the potential source regions in Shandong,
Fig. S4 plots the concentrations of $NO_2$ and $SO_2$ in different urban areas of Shandong province where
the trajectories had passed through during the same period. The concentrations of $NO_X$ and $SO_2$ in
Shandong province ranged from 31 to 78 μg m$^{-3}$ and 13 to 56 μg m$^{-3}$, respectively. The mean values of
$NO_X$ and $SO_2$ were 56 μg m$^{-3}$ and 32 μg m$^{-3}$, much higher than those of 14 μg m$^{-3}$ and 8 μg m$^{-3}$ in
Hangzhou. Hence, the air masses originating from Shandong province should be contributable to the
observed high values of aerosol secondary components in the morning of 4 September. However, it is



difficult to determine that whether the high concentrations of SNA were dominated by local atmospheric
processing or directly transported from the upstream areas. Here, we calculated the time-series of sulfur
oxidation ratio (SOR) and nitrogen oxidation ratio (NOR) as shown in Fig. 7b. The NOR and SOR in
this study are calculated as molar fraction by the following equations:
$$SOR = \frac{nSO_4^{2-}}{(nSO_4^{2-} + nSO_2)} \tag{4}$$
$$NOR = \frac{nNO_3^-}{(nNO_3^- + nSO_2)} \tag{5}$$
Both SOR and NOR had obvious increases in the morning of 4 September. Of which NOR increased
dramatically from a mean value of 0.06 from 0:00 LT on 1 August to 23:00 LT on 3 September to a peak
value of 0.52 on 04:00AM LT on 4 September. We do not think a 9-fold increase of NOR within 5 hours
was due to the local atmospheric processing. Instead, the massive input of the secondary aerosols via
long-range transport should be the major cause of the abrupt increase of SOR and NOR. It has been
recognized that secondary formation from the oxidation of $NO_X$ and $SO_2$ can occur in air masses during
the transport and directly resulted in rapid increase of $PM_{2.5}$ (Li et al., 2015). After this short particulate
pollution episode, the concentrations of SNA, OC, and EC decreased quickly in the afternoon of
September 4, demonstrating the effectiveness of emission control measures during the G20 Summit
period. In addition, all those pollutants remained at low levels throughout S4, further manifesting the
positive impact on PM reduction caused by emission control strategies.
**3.4.5. Rebounce of air pollutants in S5**
After the lifting of emission control measures, the concentrations of all the air pollutants quickly climbed,
demonstrating an abrupt worsening of air quality after the G20 Summit (Fig. 5). Mean concentrations of
SNA, OC, and EC increased significantly compared to the control stages. In detail, SNA increased 62 %,
52 %, and 37 % compared to S2-S4, with an average rise of 50 %. OC increased 45 %, 30 %, and 50 %
compared to the three stages above, with an average rise of 42 %. As for EC, the increments reached
40 %, 18 %, and 29 % with the mean value of 29 %. The substantial increases of all the air pollutants in
the post-control period further corroborated the prominent effect of emission controls on PM reduction
during the control period. As described in Sect. 3.1, the meteorological conditions in this stage were
characterized of high RH at 46-94 %, low wind speed at 0.05-2.5 m s$^{-1}$ and low radiation at 3-672 W m$^-$





$^2$. This suggested that the unfavorable meteorological conditions during S5 should additionally contribute
to the deterioration of air quality. Actually, the high concentrations of SNA, OC, and EC were mostly
observed at nighttime, accompanied with high RH and low wind speed, elucidating the important role of
meteorological conditions in the rise of particulate matters in S5. 72-hours air mass backward trajectory
clustering results illustrated that about 43 % of the trajectories travelled relative short distances, which
were restrained within the Yangtze River Delta, while the rest of the trajectories were derived from much
farther regions (Fig. 8a). This indicated that external transport should contribute almost half of the S5
periods from the perspective of synoptic meteorology. However, the CWT results (Fig. 8b-8e) showed
that the major potential sources of sulfate and nitrate with their gaseous precursors were mainly
dominated by local and regional emissions with highest hotspots around the Hangzhou Bay region. It
should be noted that a large number of the hotspots also appeared over the East China Sea as indicated
in Fig. 8b-8e. As for $NO_X$ and $NO_3^-$ (Fig. 8b & 8d), it could be visualized that the plumes over the ocean
were linked back to the hotspots over land, specifically from the southern part of Jiangsu province,
indicating the continental outflows were influential on the high levels of $NO_3^-$ during the post-G20 period.
While for $SO_2$ and $SO_4^{2-}$ (Fig. 8c & 8e), it could be seen that the hotspots over the ocean were
disconnected from the continental outflows. Emissions from ship activities are major sources of $SO_2$ over
the ocean (Fan et al., 2016;Liu et al., 2016b), which may contribute to the increase of sulfate to some
extent during the post-G20 period.

### 3.5 Formation of secondary aerosols


### 3.5.1. Secondary inorganic aerosols


The formation pathways of sulfate and nitrate were usually dominated by heterogeneous reactions as
indicated by previous studies that both these two species showed strong dependence on relative humidity
(Cheng et al., 2014;Pan et al., 2016). However, this study showed contrasting results to those previous
studies. Fig.9a & 9b plot the variations of NOR and SOR as a function of RH colored by temperature. In
addition, the relationship between NOR (SOR) and RH was investigated by grouping RH into eight bins
with an increment of 10 %. As shown in Fig. 9a, NOR was low and fluctuated within a relatively narrow
range under low RH conditions (RH<60 %). It is usually recognized that the conversion efficiency from



NO$_x$ to NO$_3^-$ via aqueous pathway was relatively low under low RH conditions. Besides, the low RH
hours were generally associated with high temperature as indicated by the colored scatters in Fig. 9a.
Nitrate was unstable and easy to decompose under high temperature, thus also resulting in low NOR
values. As RH increased (RH>60 %), NOR started to quickly increase with the decrease of temperature.
In accordance with previous studies, the variation of NOR as function of RH exhibited an exponential
growth, manifesting the heterogeneous formation of nitrate. In comparison, the variation of SOR as a
function of RH and temperature was totally different from that of NOR (Fig. 9b). The values of SOR
fluctuated much more significantly than NOR under almost all the RH conditions, showing ambiguous
relationship between SOR and RH. This relationship was further evaluated by grouping all the data into
daytime and nighttime. In the daytime, SOR showed an increasing trend with the increase of RH under
low RH conditions (RH<50 %), while it showed a slightly decreasing trend as RH increased and reached
the lowest under RH > 90 %. As discussed above, the low RH periods were mostly associated with high
temperature, which often meant strong radiation as shown in Fig. 2. This was beneficial for generating
sufficient hydroxyl radicals and promoting the subsequent photochemical reactions of sulfate formation
(Canty, 2002;Matthijsen et al., 1998). While under high RH conditions, the temperature was much lower,
which was not favorable for the photochemical formation of sulfate. This suggested the importance of
photochemical formation pathway of sulfate during the whole study period in Hangzhou. Different from
daytime, SOR showed an increasing trend with RH in nighttime under the full range of RH conditions,
indicating the aqueous processing was also crucial for the formation of sulfate.

The mean values of NOR and SOR in each of the five stages were also shown in Fig. 9a & b. Variations

of the staged SOR and NOR showed totally different patterns. The mean values of NOR remained low
in the first three stages. However, it increased to high levels in S4 and S5 due to the changed
meteorological conditions and the influence of regional/long-range transport. Variation of NOR among
the five stages showed a wide range of 0.05-0.15 with a gap of 0.1. In contrast, the SOR values among
the five stages varied weakly from 0.22-0.29, suggesting nitrate was more influenced by emissions and
the extent of long-range transport than sulfate.

Fig. 9c further shows the relationship between sulfate and nitrate as a function of RH. It is clearly

shown that high RH episodes tended to accompany with high nitrate concentrations, whereas a number
of high sulfate values appeared during low RH periods. This is quite different from the results observed
during the severe haze episodes in Beijing that high levels of both nitrate and sulfate occurred under high



RH conditions (Sun et al., 2013;Wang et al., 2016). In Fig. 9d, we also investigated the behavior of $SO_2$
and $NO_X$, the precursors of sulfate and nitrate under the computed RH bins. It is found that $SO_2$
concentrations showed a substantial decrease, while $NO_X$ concentrations increased with increasing
relative humidity, suggesting the emissions of sulfate and nitrate precursors also have a great impact on
the secondary aerosols formation in addition to the meteorological conditions. Due to the relative low
concentration of $SO_2$ under high RH conditions as well as the moderate level of SOR, the low sulfate
concentrations were expected as discussed above.
Fig. 9e shows the mass fraction of SNA in $PM_{2.5}$ as a function of RH colored by temperature. The sizes
of the filled circles corresponded to the mass concentrations of $PM_{2.5}$. Generally, the ratios of $SNA/PM_{2.5}$
increased with the elevated RH, demonstrating the significant enhancement of SNA formation under high
RH conditions. An exception should be noted that in the RH bin of 90-100 %, the $SNA/PM_{2.5}$ ratio
increased to an abnormally high value of 0.65. All data in this RH bin comes from 7 September, which
was a rainy day with accumulated precipitation of 9.5 mm (Fig. S1). Accordingly, the $PM_{2.5}$
concentrations averaged within this RH bin were the lowest according to the size of the filled circles as
shown in Fig. 9e. Thus, this data point was excluded in the following discussion. Compared to the study
in Beijing (Wu et al., 2018) which showed more obvious increase of the $SNA/PM_{2.5}$ ratio from 24 % to
55 % during the average RH from 15 % to 83 %, this study showed weaker increase of the $SNA/PM_{2.5}$
ratio from 23 % to 43 % in the similar RH range. This could be partly attributed to that the formation of
sulfate was not very sensitive to RH as shown in Fig. 9b. In addition, our results showed that the mass
concentrations of $PM_{2.5}$ didn't present an increasing trend as RH increased, which was different from
(Wu et al., 2018). In detail, (Wu et al., 2018) found significant increases of $PM_{2.5}$ concentrations from an
average of 39.4 $\mu g\ m^{-3}$ under RH < 20 % to 98.7 $\mu g\ m^{-3}$ under RH within 60-70 %, suggesting a feedback
mechanism between the aerosol liquid water and uptake of inorganic matters. Fig. 9e shows the highest
$PM_{2.5}$ concentrations occurred under the medium levels of RH, e.g. 40-60 % but not necessary under
high RH conditions. It should be noted that the average $PM_{2.5}$ concentration in this study was 39.3 $\mu g\ m^{-}$
$^3$, much lower than that of Beijing due to less strong emission intensities. In this regard, the level of $PM_{2.5}$
over the study region should be vulnerable to the inputs of outside air pollutants, especially during the
emission control period. As discussed in Sect. 3.4, regional and long-range transport were ubiquitous
during the study period. For instance, S3 was found strongly related to the long-range transport but RH





466 was the lowest among all stages. Hence, the relationship between $PM_{2.5}$ concentrations and RH was

467 ambiguous, which was attributed to the net effects of regional/long-range transport and emission control.

468 **3.5.2. Secondary organic aerosols**

469 The average OC/EC ratios were 4.0, 3.6, 4.2, 4.2, and 4.5 during the five stages with the average value

470 of 4.1. It has been recognized that if OC/EC ratios exceeding 2.0, there is production of secondary organic

471 aerosol (Cao et al., 2013). Hence, the mean ratio of OC/EC in this study implied the substantial formation

472 of SOC during the whole study, which was ascribed to the humid and warm weather conditions in the

473 summer and autumn of Hangzhou.

474 As introduced in Sect. 2.3.3, the EC-tracer method derived the $(OC/EC)_{pri}$ values in the range of

475 1.7 to 2.9 (Fig. S5), which were within or slightly higher than the $(OC/EC)_{pri}$ values of (1.15-1.85)

476 derived by (Wu et al., 2016). Due to the implementation of different control strategies, the $(OC/EC)_{pri}$

477 values fluctuated greatly among the five stages as shown in Table 1.

478 The average SOC concentrations were estimated to be 3.8, 2.2, 2.0, 1.8, and 2.2 µg m$^{-3}$ from S1 to S5,

479 respectively. The highest SOC concentration along with the highest SOC/OC ratio (0.5) in S1 can be

480 partly explained by the possibly high abundance of SOC precursors before implementation of the intense

481 emission control measures. In addition, the highest temperature and solar radiation in S1 should be also

482 responsible for the strong formation of SOC (Fig. S2 in Sect. 3.2). During S2-S4, the concentrations of

483 SOC evidently decreased compared to S1 while those of POC stayed at similar levels as S1, thus resulting

484 in obvious decrease of the SOC/OC ratios. This should be mainly ascribed to the abatement of SOC

485 precursors, e.g. VOCs. The restrictions on vehicle stocks, construction works, and giving local residents

486 extra holiday should greatly reduce the VOC emissions from vehicles, painting, residential and restaurant

487 cooking, etc. The lowest concentrations of SOC during the intense control stages reflected the

488 effectiveness of the emission control measures on suppressing the formation of secondary organic

489 aerosols. During S5, SOC had a slight rebounce compared to the control stages while POC increased

490 substantially to an average of 6.2 µg m$^{-3}$, about 50-150 % higher than the previous stages. As a result,

491 the extremely low SOC/OC (0.22), namely the very high POC/OC ratio (0.78) was estimated. This

492 suggested the primary carbonaceous emissions were greatly enhanced after the lifting of various emission

493 control measures. Allowance of all types of vehicles after the G20 Summit should be the major factor



contributing to the elevated primary carbonaceous aerosols. In addition, recovery of industries and
construction works should be also partly responsible for this.
Table S1 summarizes the SOC/OC ratios in different urban areas. Generally, the average SOC/OC
ratios in this study were lower than previous studies. Compared to the previous studies in Hangzhou, (Li
et al., 2017) estimated a SOC/OC ratio of around 40 %, slightly higher than this study. However, the
study by (Li et al., 2017) was conducted in winter, thus the SOC/OC ratio should be considered as a
lower limit. Compared to the results in (Jiao and Qi, 2007), the SOC/OC ratio in summer was 45.8 %,
much higher than this study. All the results above indicated the intense emission control measures had
exerted significantly negative impacts on the formation of secondary organic aerosols.
**3.6 Diagnose the effect of regional/long-range transport**
Fig. 10 shows the relationship between hourly sulfate/nitrate and EC during different time periods. Three
periods were defined, of which Fig. 10a & 10b consisted of all the data by excluding the regional and
long-range transport episodes as identified in the earlier discussions. It was obviously shown that
sulfate/nitrate and EC were weakly correlated. This is expected as EC is a primary particulate pollutant
emitted from incomplete combustion while sulfate and nitrate are formed from secondary reactions. As
a comparison, sulfate and EC exhibited a moderate correlation ($r^2 = 0.40$) during S3 (Fig. 10c), which
was a period identified with intensive long-range transport (Sect. 3.4.3). This phenomenon was more
evident in the quick pollution episode in S4 (Fig. 10d). Sulfate showed significant correlation with EC
($r^2 = 0.67$) and a moderate correlation was also observed between nitrate and EC ($r^2 = 0.40$). This
"abnormally" positive correlation between species that were derived from different sources and
formation pathways indicated that the temporal variations of aerosol components were dominated by
physical processes rather than atmospheric chemical processing. That is to say, it was driven by the
transport which brought massive inputs of air pollutants, and then diluted or accumulated synchronously.
Hence, to assess whether there is a significant correlation between EC and secondary aerosol components
could possibly judge the occurrence and extent of regional and long-range transport.
Fig. 11 further evaluated the effect of regional/long-range transport on the extent of the formation of
secondary inorganic aerosols (i.e. the SNA/EC ratio) and $PM_{2.5}$ levels by grouping each stage to a wind
direction bin of 45 degrees. All data were colored by wind speed and the sizes of the filled circles



corresponded to the PM$_{2.5}$ concentrations. It is clearly shown that in S1 both the SNA/EC ratios and PM$_{2.5}$
concentrations became relatively high from the northeast and northwest, i.e. the upstream polluted
regions with much higher emission intensities than the Yangtze River Delta. Statistically, the SNA/EC
ratios from the wind sector of northwest to northeast were moderately higher (1.4-fold) than from the
other directions, suggesting the regional transport was not prominent. S2 generally exhibited a similar
pattern as S1. The difference was the overall decrease of PM$_{2.5}$ concentrations from all the wind sectors.
This was partly ascribed to the emission control and was also related to the higher wind speed as
visualized by the colored circles.

Compared to S1and S2, S3 showed an opposite pattern of SNA/EC as a function of wind direction.

The relatively high SNA/EC ratios were observed from the northeast to southwest as well as for the PM$_{2.5}$
concentrations. This was consistent with the CWT results that the southern areas of Hangzhou were the
potential source regions of high sulfate and nitrate (Sect. 3.4.3). It should be noted that although regional
transport was observed during this stage, the SNA/EC ratios from the northeast to southwest were only
18-27 % higher than the other directions, much lower than the regional transport from the north as
discussed below. We think this was due to that the southern part of the Yangtze River Delta had lower
emission intensities than the north, thus limiting the elevation of the SNA/EC ratios during the transport.

In S4, a distinctly different pattern of SNA/EC from the other stages showed the extremely high ratios

of SNA/EC from the north and northeast, about 2.4-3.4 times that from the other directions. This
corroborated with the air mass backward trajectory analysis in Fig. 7, verifying the transport path from
continental outflow via sea breeze. The high ratios of SNA/EC from the north and northeast in S4 were
almost 4 times higher than the other four stages, which was also consistent with the discussions in Sect.
3.4.4 that SOR and NOR abruptly increased during the quick pollution episode in the morning of 4
September. Specifically, the two highest SNA/EC ratios were accompanied with large error bars,
suggesting the great fluctuations of SNA/EC in the divided wind direction intervals. This was related to
the characteristics of the sea breeze from the north to the east. In most circumstances, the sea breeze
directly from the ocean exerted cleansing effect, lowering the levels of air pollutants. However, the land
sea breeze in S4 could have transported abundant air pollutants back to the land, worsening the air quality
in this study. Thus, the PM$_{2.5}$ levels associated with the sea breeze could have been in a wide range, thus
generating large error bars.



As for S5, the pattern of SNA/EC as a function of wind direction was somewhat as similar as S1 and
S2. The values of SNA/EC were much higher than those of S1-S3 in almost all the wind direction
intervals. Lifting of emission control measures should be the major cause. In addition, unfavorable
meteorological conditions (e.g. low wind speed for the high SNA/EC groups) also accelerated the
formation and accumulation of secondary aerosols.
**4 Conclusions**
In this study, atmospheric chemical compositions from 15 August to 12 September before, during, and
after the 2016 Hangzhou G20 Summit were monitored. Water-soluble ions, organic/elemental carbon,
and gaseous pollutants were continuously measured. Soluble ions and carbonaceous matters are the major
components of fine particles, accounting for 60-80 % of $PM_{2.5}$. The average $PM_{2.5}$ concentrations during
the five defined stages (one reference stage, three control stages, and one post-G20 stage) were 37.4,
31.8, 40.4, 35.0, and 49.5 $\mu g\ m^{-3}$, respectively. In general, the emission control measures were effective
in lowering the concentrations of fine particles. The impact of emission control measures on perturbing
the air quality was fully assessed. The major findings are summarized as below:
1.     Both sulfate and nitrate showed dependence on RH, but RH played a more important role in the
formation of nitrate. In addition, the formation of sulfate was found highly related to the
photochemical reactions, especially during daytime. This is different from previous studies on haze
in Beijing that the formation of sulfate was more influenced by RH.
2.     Air mass backward trajectory and CWT analysis suggested that regional/long-range transport were
ubiquitous even during the strict vehicle stock control period. Long-range transport from upstream
regions such as Shandong and Jiangsu was diagnosed as the main cause of high NOx concentrations.
3.     One high particulate pollution episode observed in the morning of 4 September (the first day of the
G20 Summit) was found related to the continental outflow via the sea-to-land breeze. Abrupt
increases of SOR and NOR values were observed during this short pollution episode, especially for
NOR with a 9-fold increase within 5 hours. Local atmospheric processing in Hangzhou shouldn't
be the driving force. Instead, the formation of secondary aerosols in the humid sea breeze or direct
inputs of secondary aerosols from upstream source regions were responsible for this most severe
particulate pollution during the study period.





4.      The concentrations of estimated SOC showed significant decreases during all the control stages.

Specifically, the SOC diurnal pattern was modified and its peaks in the daytime were greatly

reduced, indicating the influence of emission control effects on the SOC formation.

This study shows that the various emissions control measures implemented for the Hangzhou G20
Summit indeed had a positive impact on the reductions of aerosol concentrations in a short period of
time. However, the regional/long-range transport may offset the local emission control effects to some
extent. Finally, the post-G20 period showed a quick and sustained deterioration of air quality, which was
as similar as the 2010 Shanghai Expo and 2014 Beijing APEC when all the emission control measures
were lifted.
**Acknowledgments**
This work was financially supported by the National Key Research and Development Program of China,
National Natural Science Foundation of China (91644105, 41429501) and Hangzhou Science and
Technology Development Plan (20160533B83, 20172016A07).

*Data availability.* All data used in this paper are available by contacting Kan Huang
(huangkan@fudan.edu.cn).

*Competing interests.* The authors declare that they have no conflict of interest.

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





**Table 1.** The ratios of mean $(OC/EC)_{pri}$, SOC/OC, and POC/OC and the mass concentrations
($\mu g\ m^{-3}$) of OC, EC, POC, and SOC during the five stages.


| Stages | $(OC/EC)_{pri}$ | EC | POC | SOC | SOC/OC | POC/OC | OC/EC |
|---|---|---|---|---|---|---|---|
| S1 | 1.7 | 1.8 | 3.0 | 3.8 | 0.50 | 0.50 | 4.0 |
| S2 | 1.8 | 1.4 | 2.4 | 2.2 | 0.41 | 0.59 | 3.6 |
| S3 | 2.6 | 1.4 | 3.8 | 2.0 | 0.34 | 0.66 | 4.2 |
| S4 | 1.8 | 1.4 | 2.4 | 1.8 | 0.44 | 0.56 | 4.2 |
| S5 | 2.9 | 2.2 | 6.2 | 2.2 | 0.22 | 0.78 | 4.5 |
| Mean | 2.20 | 1.7 | 3.7 | 2.6 | 0.38 | 0.62 | 4.1 |






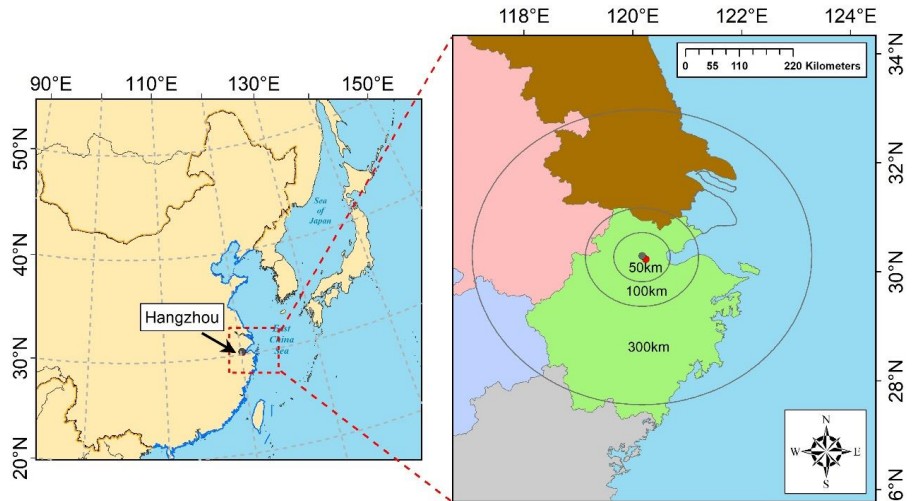





**Figure 1.** Geographic locations of Zhejiang (green), Shanghai (blue), Jiangsu (brown), Anhui
(pink), Jiangxi (purple), and Fujian (gray) as visualized by different colors. The red dot in the right
panel represents the main venue of the Hangzhou G20 Summit and the gray one denotes the
location of the observational site in this study. By taking the main venue as the center of the
emission control zone, three regions were set up with respect to different control intensities, i.e.
the core emission control zone (r <50km), the strict emission control zone (r <100km), and the
general emission control zone (r <300km).
















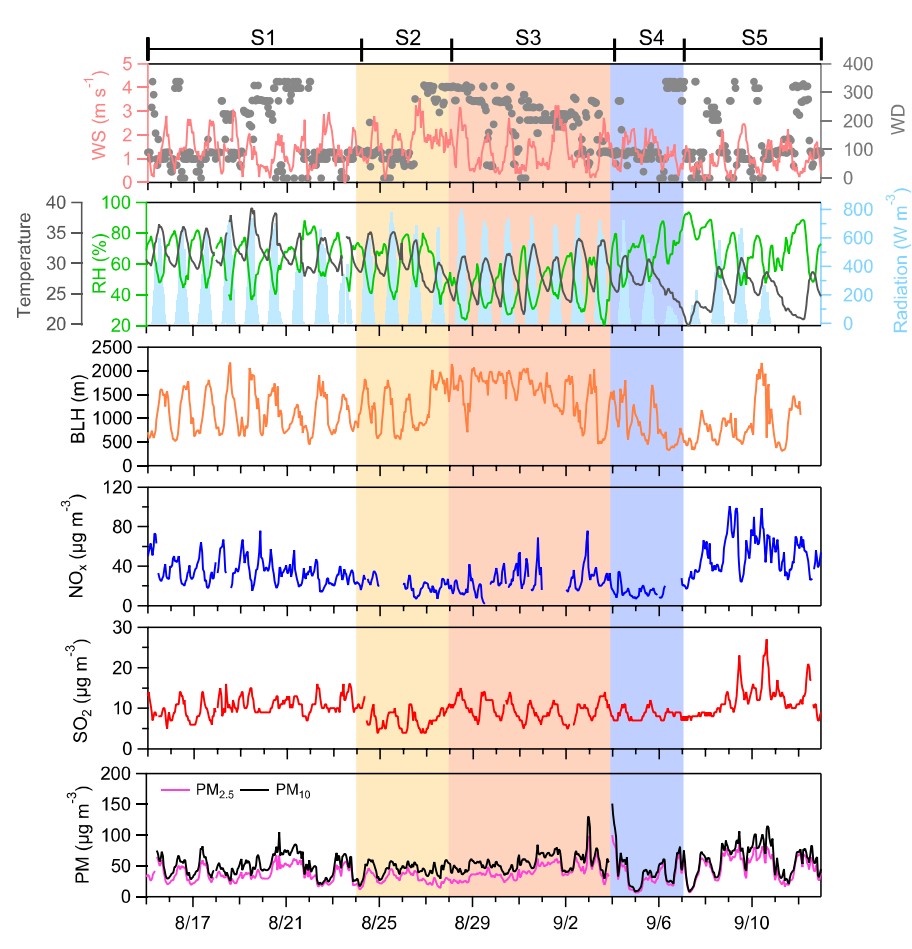

**Figure 2.** Time series of hourly PM$_{2.5}$ and PM$_{10}$ concentrations, together with trace gases

(NO$_x$, SO$_2$) and meteorological parameters (Wind Speed (WS), Wind Direction (WD), Boundary

Layer Height (BLH), Relative humidity (RH), Temperature (T), and Radiation). The defined five

stages from S1-S5 are marked on the top of the figure.







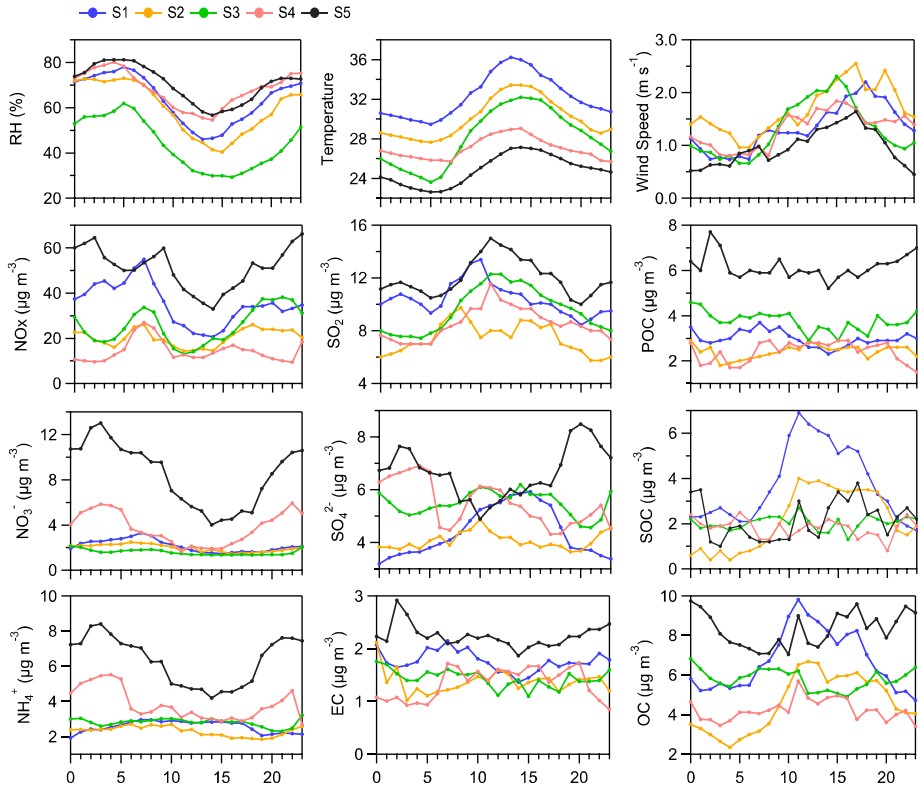



**Figure. 3.** Diurnal profiles of PM$_{2.5}$ species, gaseous pollutants, and meteorological variables
during the five stages.
















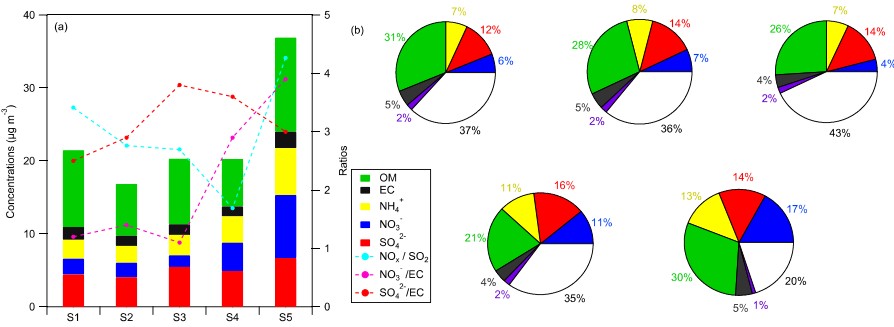



**Figure 4.** (a) Mean concentrations of major chemical components of PM$_{2.5}$ with respect to
different stages. OM (organic matter) was estimated based on OC multiplied by a factor of 1.8 in
this study (Xing et al., 2013). The mass ratios of NO$_x$/SO$_2$, NO$_3^-$/EC, and SO$_4^{2-}$/EC at each stage
are also plotted. (b) Mass fractions of the measured aerosol chemical components from S1 to S5.
The different color numbers for the pie chart denote the mass fractions of major aerosol
constituents, i.e., green for organic matter (OM), black for elemental carbon (EC), red for SO$_4^{2-}$,
dark blue for NO$_3^-$, yellow for NH$_4^+$, purple for the sum of Ca$^{2+}$, Mg$^{2+}$, K$^+$, Na$^+$, and Cl$^-$.























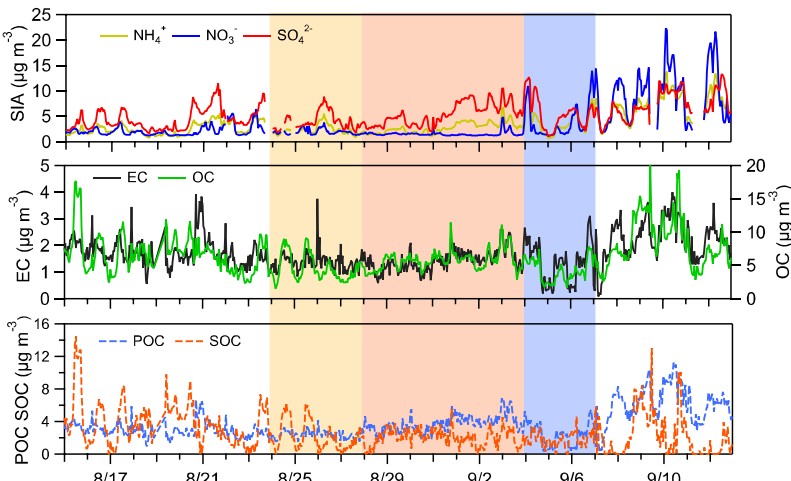



**Figure 5.** Time-series of PM$_{2.5}$ chemical components during the entire study period.























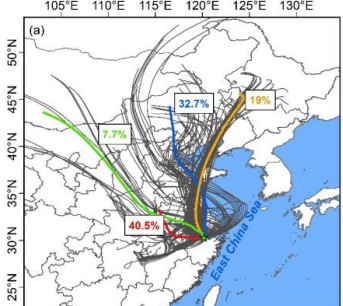


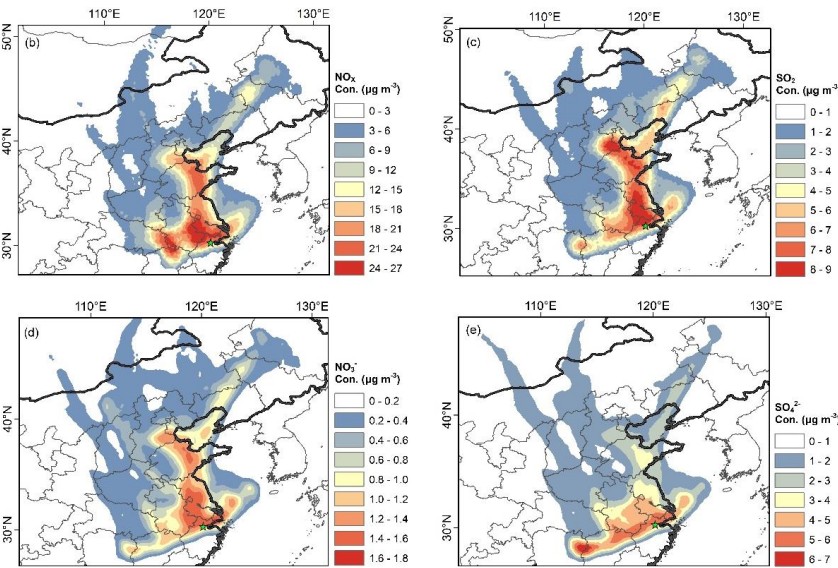


**Figure 6.** (a) Cluster analysis of the 72-h air mass backward trajectories starting at 500m in
Hangzhou during S3. Concentration-Weighted Trajectory (CWT) maps for (b) $NO_x$, (c) $SO_2$, (d)
$NO_3^-$, and (e) $SO_4^{2-}$ for the whole S3 period. The location of the monitoring site is marked by a
green star.














**Figure 7.** (a) Time series of hourly concentrations of SNA, Cl⁻, and meteorological
parameters (WS, WD, RH, and Radiation) during S4; (b) 48-h air mass backward trajectories for
the short pollution episode in the morning of September 4, 2016; (c) Hourly variations of SOR and
NOR during the entire study period. The highlighted period represents the short pollution episode
in the morning of September 4, 2016.



















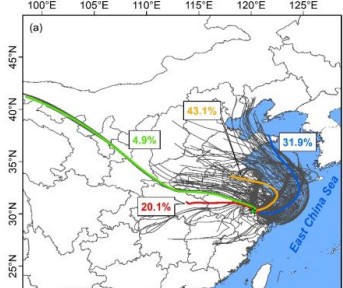


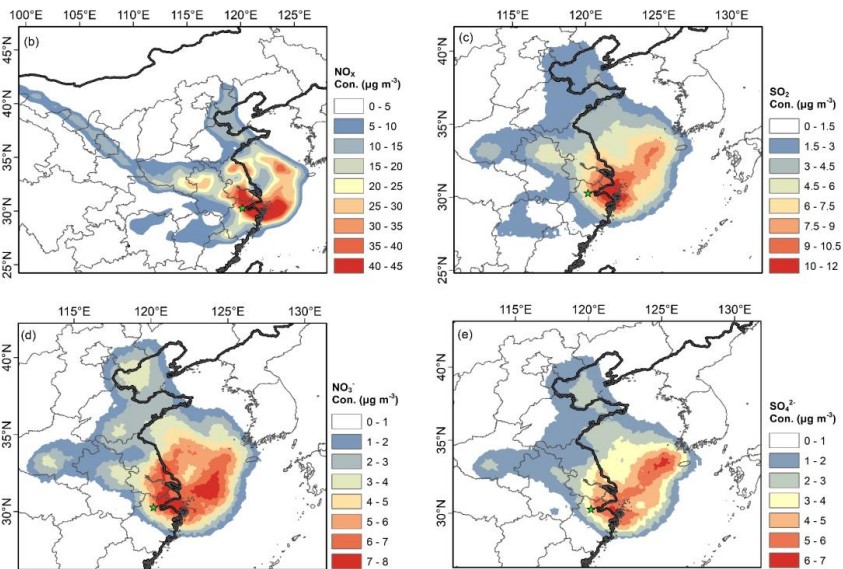


**Figure 8.** (a) Clustering analysis of 72-h air mass backward trajectories starting at 200m
during S5. The choose of 200m is due to the low boundary layer height (about 500m on average)
during this stage. Concentration-Weighted Trajectory (CWT) maps for (b) $NO_x$, (c) $SO_2$, (d) $NO_3^-$,
and (e) $SO_4^{2-}$ for the whole S5 period. The location of the monitoring site is marked by a green
star.





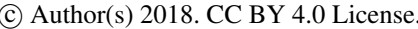




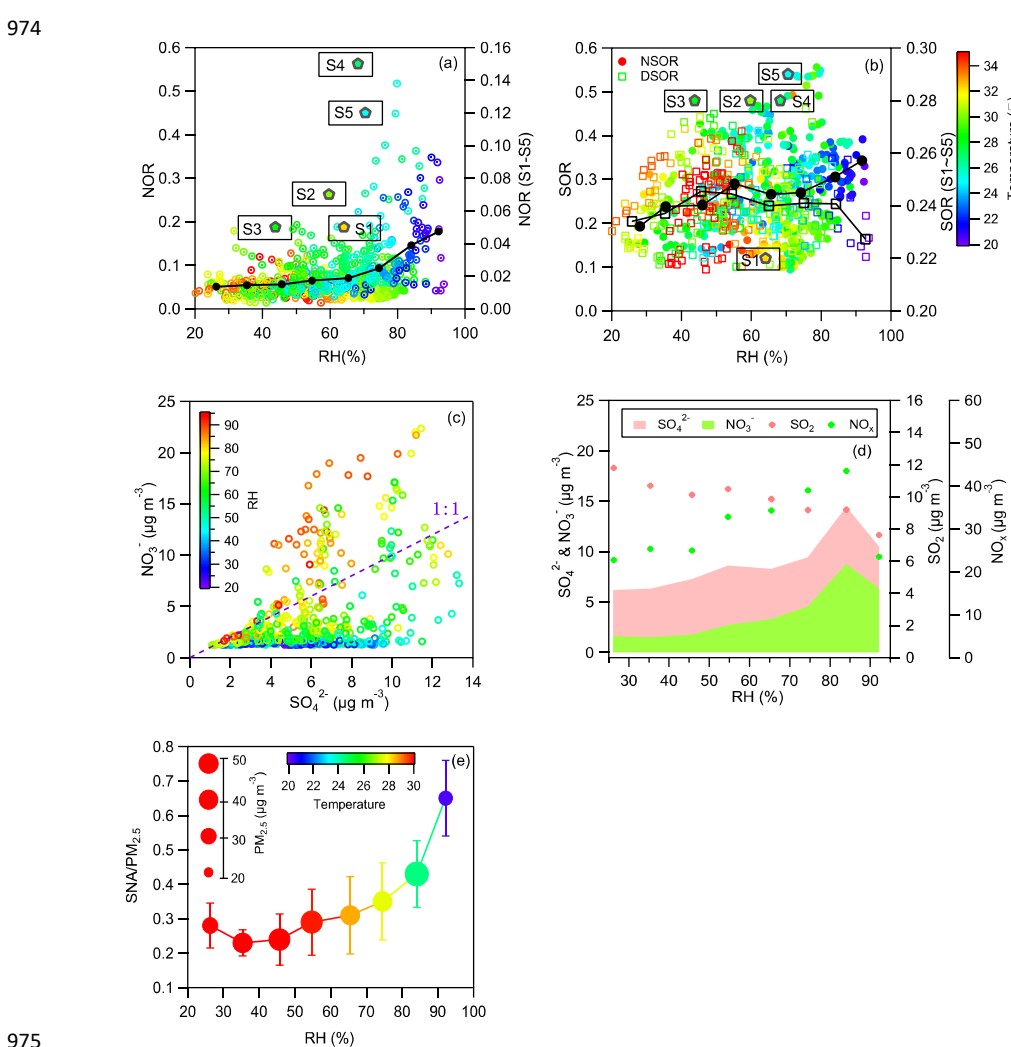



**Figure 9.** Hourly nitrogen oxidation ratio (NOR) (a) and sulfur oxidation ratio (SOR) (b)
plotted against RH colored with temperature. The pentagons in (a) & (b) denote the mean values
of NOR and SOR in each stage and the values use the right axis; DSOR and NSOR mean the SOR
values during daytime and nighttime, respectively. (c) Relationship between hourly sulfate and
nitrate colored by RH. (d) Hourly $SO_4^{2-}$, $NO_3^-$, $SO_2$, and $NO_x$ as a function of RH. (e) The ratio of
SNA/PM$_{2.5}$ as function of RH in each bin of 10%. The filled circles are colored with temperature
and the sizes of the circles correspond to the mass concentrations of PM$_{2.5}$.









**Figure 10.** Hourly sulfate (a) and nitrate (b) plotted against EC colored with $PM_{2.5}$ mass
concentrations. Data in Fig. 10a & 10b included the whole study period by excluding data in Fig.
10c & 10d. (c) Relationship between hourly sulfate, nitrate and EC from 0:00 AM on 28 August to
21:00 PM on 3 September. (d) Relationship between hourly sulfate, nitrate and EC from 22:00 PM
on 3 September-5:00 AM LT on 4 September, respectively.














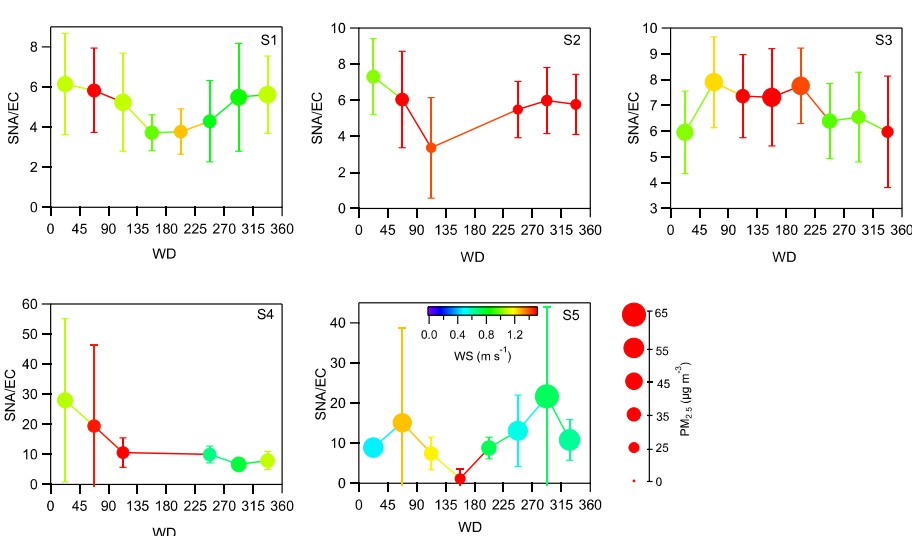



**Figure 11.** Variation of the ratio of SNA/$PM_{2.5}$ in eight wind direction sectors with the bin

width of 45 degrees during the five stages. The filled circles are colored with wind speed and the

sizes of the circles correspond to the mass concentrations of $PM_{2.5}$.
