# Peer review of "Counteractive effects of regional transport and emissions control on the formation of fine particles: a case study during the Hangzhou G20 Summit"

_Atmospheric Chemistry and Physics, 2018_

## Referee Comment (RC1) · Anonymous Referee #1 · 16 Jul 2018

The manuscript titled "Counteractive effects of regional transport and emissions control on the formation of fine particles: a case study during the Hangzhou G20 Summit" by Ji et al. gave a detailed analysis of the major chemical components in particulate matters during the 2016 Hangzhou G20 mega-event. This mega-event has attracted substantial attentions as emission controls had been implemented at both large-scale areas and with significant intensities firstly over the Yangtze River Delta region. However, the effectiveness has been rarely reported. The study by Ji et al. comprehensively investigated the chemical characteristics of fine particles during the five defined stages. It is found that particle pollution during some control periods are even higher than the pre-control period. Regional and long-range transport are found playing an important

role. It is highlighted that the formation of secondary aerosols is quite different from previous studies in northern China, e.g. the dependence of sulfate formation on relative humidity, SNA/PM2.5 ratios as a function of humidity, etc. It is also found that the SNA/EC ratios are strongly impacted by the extent of long-range transport.

Overall, this is an interesting and important work for understanding the relative importance of emission control and long-range transport. The interpretation of the results is generally well described. I recommend it for publication with minor revisions as listed below.

Major Comments: Line 251 - 267, Why S2 didn't show absence of SOC peaks during daytime as S3 and S4? S2 - S4 are all among the emission control periods. Line 329 - 331, I didn't see sulfate from Hunan in the figure. What's the reason of the different potential source regions of SO2 and sulfate? Line 394 - 397, high sulfate hotspots are seen in Figure 8 and it is concluded that shipping activities could be the cause. How is this conclusion supported? Line 444 - 467, The results of the SNA/PM2.5 ratios and PM2.5 concentrations as a function of relative humidity are overall interesting and showed differently from previous results. This suggested that the formation mechanisms of secondary aerosol are different from the north to south in China. However, the conclusion "Hence, the relationship between PM2.5 concentrations and RH was ambiguous, which was attributed to the net effects of regional/long-range transport and emission control" was not well supported. The authors may more statistically explain it.

Minor Comments: Line 187, change "first" to "firstly" Line 188, add "value" after (OC/EC)pri Line 202, it is unclear what's the "most stringent emission control stage", please add more details. Line 206, how is the boundary height measured? Line 241, change "about" to "from" Line 246, change "7 September and 9 September" to "September 7th and 9th" Line 272 - 273ïijŇDoes this sentence refer to Fig. 4a? If so, the trace ions (Na+, K+, Ca2+, Mg2+, and Cl-) are not included. Line 276, add "compared to S1" after "from S2 to S4". Line 284, compared to the average of S2-S4?

[Figure]

Please write clearly. Line 296, change "The NO3-/EC variation" to "The variation of NO3-/EC" Line 304, it should be "well developed boundary layer" but not "well developed boundary layer height" Line 305, "be subjected to" should be "be subject to". Line 358, SO2 in the equation is a typo. Line 374, add "respectively" after "compared to S2 - S4" Line 441, change "relative" to "relatively" Line 525, change "sector" to "sectors" Line 531, it should be written clearly "from the northeast to southwest" is clockwise or anti-clockwise. Line 549 - 550, I guess the error bars refer to the SNA/EC ratio but not PM2.5.

Technical Comments: Figure 1. the right panel of the figure is suggested to use a terrain shapefile. Figure 2. Add units for wind direction and change its scale to 0~360; Add unit for temperature; The missing data in the figure should be explained; The definitions of S1- S5 are better added into the caption. Figure 3. Add unit for temperature; Add title for the X-axis; Figure 4. The dotted lines are not easy to see; Enlarge the titles of the Y axis; Mark S1 - S5 on the pie charts. Figure 5. The size of the Y-axis for the three panels seems different, make it consistent; Indicate five stages at the top of the figure. Figure 6. The geographic areas in the back trajectory map are different from the PSCF maps. Make it consistent as well as for Figure 8. Figure 7. Add unit for wind direction and change its scale to 0~360; In Fig. 7c, the unit of the trajectory height is also missing. Figure 9. Add the unit of temperature; Add the description of the error bar in the caption Figure 11. Add the unit of wind direction; Add the description of the error bar in the caption

---

## Referee Comment (RC2) · Anonymous Referee #2 · 30 Jul 2018

Review of the manuscript entitled"Counteractive effects of regional transport and emissions control on the formation of fine particles: a case study during the Hangzhou G20 Summit"by Ying Ji et al. submitted for possible publication in the ACP

Comments: The subject is appropriate to ACP. This manuscript presents the results of an intensive field campaign conducted with focus on aerosol chemistry and gaseous precursors from 15 August to 12 September, 2016, to evaluate the effect of temporary emissions control measures on air quality during the 2016 G20 Summit held in Hangzhou, China. The results show that the concentrations of fine particles were reduced during the intense emission control stages mainly because of the decreases

of secondary organic aerosols via the suppression of daytime peak SOC formation. They also found that the effect of long-range transport on the air quality of Hangzhou was ubiquitous and that unexpectedly high NOx concentrations were observed during the control stage when the strictest restriction on vehicles was implemented, owing to the contribution from upstream populous regions such as Jiangsu and Shandong provinces. The results are interesting and useful. Therefore, I recommend clearly the acceptance for publication of this manuscript after minor revisions. Several editorial comments for improving the information content and presentation of the paper are listed as follows: 1. L 22: It should be "contributions" instead of "contribution", 2. L52: It should be "air pollutant emissions" instead of "air pollutants emissions". There are many other English grammar errors in the text part. Please fix them when you revise your manuscript. 3. Lines 46-56: About the background and study of air pollution during the Hangzhou G20 summit, some results have been published. Regarding impact of effectiveness of short-term emission control schemes on air quality in Hangzhou during the 2016 G20 Summit, the paper of Li et al ( Li P, Wang L, Guo P, et al. High reduction of ozone and particulate matter during the 2016 G-20 summit in Hangzhou by forced emission controls of industry and traffic. Environ Chem Lett, 15:709–715, DOI 10.1007/s10311-017-0642-2) has done significant work. The authors should cite this paper. 4. Lines 95-96: Before line 96, it will be good to add a paragraph as background to summarize the current status of air pollution study in Hangzhou such as the paper of Yu S, Zhang Q, Yan R, et al. Origin of air pollution during a weekly heavy haze episode in Hangzhou, China[J]. Environmental Chemistry Letters, 2014, 12(4):543-550.

---

## Referee Comment (RC3) · Anonymous Referee #3 · 25 Aug 2018

This manuscript presents analyses of aerosol and meteorological observations taken at an urban site in Hangzhou, China, before, during, and after the G20 Summer in Sep 2016. The measurements have rich datasets of aerosol speciation, which allows for the use of species correlation to diagnose transport and source characteristics. In combination with back trajectory analysis, the authors attempted to estimate the contribution of local, regional, and continental scale transport to the observed variations in aerosol concentrations at that site. The paper is overall well-written with solid analysis. I have two major concerns about the scale the measurements would represent and robustness of trajectory analysis.

[Figure]

First, the paper did not specify the resolution of the meteorological data that drive the HYSPLIT back trajectories. On line 157-158, it says the GDAS is used. The GDAS data has a horizontal resolution of 1 degree by 1 degree, roughly 100 km x 100 km. Thus, it is possible for the GDAS-driven back trajectories to distinguish between local and regional transport. The study domain shown in Figure 1 (right), which is on the scale of 300 km at the maximum, will be covered by only a few grid boxes. This short-coming would make all the related discussion of transport patterns and source location attributions highly uncertain. I think the authors should use a much finer meteorological dataset to drive the back trajectory model, e.g. from WRF with horizontal resolutions of a few km at least.

Second, I found the sea breeze related discussion groundless. First, they do not provide concrete evidence there was indeed sea breeze circulation during the study period. Sea breeze is a localized circulation pattern on the scale of a few kilometers with distinctive diurnal reversals of winds: offshore flow during the day and onshore during the night. The authors did not establish any of these patterns in the paper. I actually doubt there exists a sea breeze circulation in Hangzhou. Second, given the sea breeze as a local circulation pattern with wind reversals, it is unlikely one can detect continental outflow from the sea breeze without careful analysis of the flow patterns or modeling. What happen more often with sea breeze is the recirculation of local pollutions, as often seen in coastal cities around the globe. Offshore winds at night and during early morning push urban pollutions to the shallow marine boundary layer, which then recirculate back to the urban area in the afternoon by the offshore flow. So, if the authors could establish there was indeed sea breeze circulation during the study period, the follow-up discussion should be restricted to the local scale circulation, rather than the continental background. The local scale focus also reinforce my first comment that a finer-resolution back trajectory analysis should be used.

---

## Author Response (AR1)

**Response to Reviewer Comments #1**

The manuscript titled "Counter active effects of regional transport and emissions control on the formation of fine particles: a case study during the Hangzhou G20 Summit "by Ji et al. gave a detailed analysis of the major chemical components in particulate matters during the 2016 Hangzhou G20 mega-event. This mega-event has attracted substantial attentions as emission controls had been implemented at both large-scale areas and with significant intensities firstly over the Yangtze River Delta region. However, the effectiveness has been rarely reported. The study by Ji et al. comprehensively investigated the chemical characteristics of fine particles during the five defined stages. It is found that particle pollution during some control periods are even higher than the pre-control period. Regional and long-range transport are found playing an important role. It is highlighted that the formation of secondary aerosols is quite different from previous studies in northern China, e.g. the dependence of sulfate formation on relative humidity, SNA/PM2.5 ratios as a function of humidity, etc. It is also found that the SNA/EC ratios are strongly impacted by the extent of long-range transport. Overall, this is an interesting and important work for understanding the relative importance of emission control and long-range transport. The interpretation of the results is generally well described. I recommend it for publication with minor revisions as listed below.

We thank the referee for his or her insightful and helpful comments and suggestions to improve the quality and clarity of the manuscript. Please find below our point-by-point reply to the specific comments. All the changes made to the manuscript were highlighted in green.

**Major Comments:**

Line 251 - 267, Why S2 didn't show absence of SOC peaks during daytime as S3 and S4? S2 - S4 are all among the emission control periods

The emission control intensities of S2-S4 were gradually enhanced, and S2 was the weakest emission control stage. Specifically, S2 didn't add the emission control on vehicles, which were important sources of VOC. Therefore, the impact of emission control measures on SOC at this stage was not as significant as S3 and S4. In addition, the meteorological conditions (e.g. temperature and radiation intensity) of this stage were beneficial to the photochemical generation of SOC.

In the revised manuscript, we have added the related description.

Line 329 - 331, I didn't see sulfate from Hunan in the figure. What's the reason of the different potential source regions of SO2 and sulfate?

We have redrawn the air mass trajectories with a finer-resolution of 0.5° × 0.5° as the third reviewer suggested. And Hunan was not the potential region of the sulfate.

[Figure]

As shown in the figure above, the SO2 CWT plot indicated hotspots mainly from Hebei, Shandong, Jiangsu, and southern Anhui, while the potential sources of sulfate were mainly ascribed to regions south of Hangzhou, i.e., the conjunction areas of Jiangxi, Anhui, and Zhejiang provinces. This should be attributed to the impact of meteorological conditions on the formation of sulfate. Table S1 shows the temperature in different areas that the hotspots had covered during the three days before S3 according to the 72h air mass back trajectory in CWT analysis. It is obviously shown that the temperature in the northern regions were much lower than in the southern regions. The high temperature in the south should be favorable for the photochemical formation of sulfate in summer. This could be the reason of the different potential source regions of $SO_2$ and sulfate.

**Table S1.** The temperature in different areas that the hotspots had covered during the three days before S3 according to the 72h air mass back trajectory in CWT analysis.

| Date | Tangshan | Langfan | Cangzhou | Lianyungang | Yangzhou | Wuxi | Huangshan | Xuancheng | Shangrao |
|------|----------|---------|----------|-------------|----------|------|-----------|-----------|----------|
| 8/25 | 23 | 25 | 23 | 26 | **28** | **29** | **30** | **29** | **32** |
| 8/26 | 21 | 22 | 21 | 23 | 24 | **27** | **29** | **27** | **32** |
| 8/27 | 22 | 22 | 22 | 23 | 24 | 25 | 25 | 24 | 27 |
| 8/28 | 22 | 24 | 22 | 24 | 24 | 25 | 26 | 24 | 25 |
| 8/29 | 23 | 23 | 23 | 25 | 23 | 25 | 24 | 24 | 25 |
| 8/30 | 23 | 24 | 24 | 24 | 23 | 25 | 24 | 23 | 25 |
| 8/31 | 24 | 26 | 25 | 24 | 24 | 25 | 24 | 25 | 24 |

Line 394 - 397, high sulfate hotspots are seen in Figure 8 and it is concluded that shipping activities could be the cause. How is this conclusion supported?

Our sampling point is located in the northeast of Hangzhou, close to the Ningbo-Zhoushan Port, which ranks as the second largest port and has the largest cargo throughput among China's top ten port. Previous studies have shown that ship emissions have an important impact on the Yangtze River Delta region and Eastern China (Fan et al., 2016;Liu et al., 2016). (Liu et al., 2016) has even shown shipping emissions could contribute 20−30% (2−7 μg/m³) of the total $PM_{2.5}$ within tens of kilometers of coastal and riverside Shanghai during ship-plume-influenced periods in spring and summer. (Fan et al., 2016) estimated the total emissions of $SO_2$ and NOx reached $3.8 \times 10^5$ and $7.1 \times 10^5$ tons/yr of the study area (119°E to 125°E and 27°N to 36°N) in 2010, respectively.

Line 444 - 467, the results of the SNA/PM$_{2.5}$ ratios and PM$_{2.5}$ concentrations as a function of relative humidity are overall interesting and showed differently from previous results. This suggested that the formation mechanisms of secondary aerosol are different from the north to south in China. However, the conclusion "Hence, the relationship between PM$_{2.5}$ concentrations and RH was ambiguous, which was attributed to the net effects of regional/long-range transport and emission control" was not well supported. The authors may more statistically explain it.

As shown in the figure below, we added the average ratios of SNA/PM$_{2.5}$ and PM$_{2.5}$ concentrations during the five stages denoted by the open circles. The size of the circles corresponded to the PM$_{2.5}$ concentrations. It was obviously shown that the positions of S4 and S5 circles largely deviated from the statistical curve between SNA/PM$_{2.5}$ and RH. This implied that the two stages were subject to other factors such as the influence of regional transport. From the analyses in the manuscript in Section 3.4, regional and long-range transport were ubiquitous during S4 and S5. Thus, it was likely that the input of transported air pollutants disturbed the relationship between PM$_{2.5}$ and RH to some extent. As for S3, although transport evidence was also revealed during this stage, the extent of long-range transport was relatively weak, which will be discussed in Section 3.6. Hence, the mean SNA/PM$_{2.5}$ ratio and PM$_{2.5}$ concentration during S3 still coincided with the statistical curve.

[Figure]

In the revised manuscript, we have added the writings above to clarify the reviewer's comment.

**Minor Comments:**
Line 187, change "first" to "firstly" Line 188, add "value" after (OC/EC)pri
Thanks for pointing this out. It is changed as suggested.

Line 202, it is unclear what's the "most stringent emission control stage", please add more details.
Thanks for the suggestion. The emergency controls on VOC and PM2.5 precursors emissions were enacted based on the air quality forecasting if a pollution day was predicated. We have explained it more clearly in the revised submission.

The height of the boundary layer was estimated based on a co-located aerosol lidar. The lidar echo slope method was used to calculate the boundary layer height. Typically, there is a thermal inversion layer at the interface between the atmospheric boundary layer and the upper free atmosphere, which traps a large amount of aerosols within the lower boundary layer. At the boundary between the atmospheric boundary layer and the free atmosphere, the aerosol extinction coefficient will undergo a quick attenuation process. Above the inversion layer, the aerosol concentration is generally low. Thus, the height of the boundary layer is determined based on the height where that the decreasing rate of aerosol extinction coefficient is most abrupt.

In the revised manuscript, we have added the description of boundary layer height into the method section.

It is changed.

It is changed as suggested.

Thanks for pointing this out. Yes, the sentence of line 272-273 refer to Fig.4a and the trace ions (Na+, K+, Ca2+, Mg2+, and Cl-) are not included. It is now corrected.

It is changed as suggested.

Thanks for pointing this out. We have revised the sentence as "After the G20 Summit, the sum of SNA, OM, and EC increased 42 %, 52 %, and 62 % compared to S2-S4, respectively, ……"

It is changed as suggested.

Thanks for pointing out this mistake. The sentence has been revised.

Thanks for pointing out this mistake. It is changed.

Thanks for pointing out this mistake. This typo has been corrected.

Line 374, add "respectively" after "compared to S2 - S4"
It is added as suggested.

Line 441, change "relative" to "relatively"
It is changed.

Line 525, change "sector" to "sectors"
It is changed.

Line 531, it should be written clearly "from the northeast to southwest" is clockwise or anti-clockwise.
We thank the reviewers for his/her comments and suggestions and we have revised the sentence into "The relatively high SNA/EC ratios were observed anti-clockwise from the northeast to southwest as well as for the PM2.5 concentrations"

Line 549 - 550, I guess the error bars refer to the SNA/EC ratio but not PM2.5.
Yes, the error bars refer to the SNA/EC

**Technical Comments:**
Figure 1. the right panel of the figure is suggested to use a terrain shape file.
We have now redrawn Figure 1 as suggested. Please check it in the revised submission.

Figure 2. Add units for wind direction and change its scale to 0~360; Add unit for temperature; The missing data in the figure should be explained; The definitions of S1- S5 are better added into the caption.
These are all added/changed as suggested. The missing data were due to the malfunction or maintenance of the instruments.

Figure 3. Add unit for temperature; Add title for the X-axis;
These are all added as suggested.

Figure 4. The dotted lines are not easy to see; Enlarge the titles of the Y axis; Mark S1 - S5 on the pie charts.
We have now re-plotted Figure 4 and have marked S1-S5 on the pie charts. Please check it in the revised submission.

Figure 5. The size of the Y-axis for the three panels seems different, make it consistent; Indicate five stages at the top of the figure.
Thanks for pointing this out. We have changed the three panels to same size in Figure 5 and the five stages were indicated at the top of the figure.

Figure 6. The geographic areas in the back trajectory map are different from the PSCF maps. Make it consistent as well as for Figure 8.

We have made them consistent both in Figure 6 and Figure 8. Please check it in the revised submission.

Figure 7. Add unit for wind direction and change its scale to 0~360;
We have re-plotted Figure 7 as suggested.

In Fig. 7c, the unit of the trajectory height is also missing.
Thanks for pointing this out. We have now added the unit of the trajectory height in Figure 7c.

Figure 9. Add the unit of temperature; Add the description of the error bar in the caption
We have re-plotted Figure 9 as suggested and the description of the error bar was added in the caption.

Figure 11. Add the unit of wind direction; Add the description of the error bar in the caption
We have re-plotted Figure 11 as suggested and the description of the error bar was added in the caption.

**Response to Reviewer Comments #2**

The subject is appropriate to ACP. This manuscript presents the results of an intensive field campaign conducted with focus on aerosol chemistry and gaseous precursors from 15 August to 12 September, 2016, to evaluate the effect of temporary emissions control measures on air quality during the 2016 G20 Summit held in Hangzhou, China. The results show that the concentrations of fine particles were reduced during the intense emission control stages mainly because of the decreases of secondary organic aerosols via the suppression of daytime peak SOC formation. They also found that the effect of long-range transport on the air quality of Hangzhou was ubiquitous and that unexpectedly high NOx concentrations were observed during the control stage when the strictest restriction on vehicles was implemented, owing to the contribution from upstream populous regions such as Jiangsu and Shandong provinces. The results are interesting and useful. Therefore, I recommend clearly the acceptance for publication of this manuscript after minor revisions. Several editorial comments for improving the information content and presentation of the paper are listed as follows.

We thank the reviewer for his/her careful reading of our manuscript and the supportive comments. We address the referee's specific comments as below. All the changes made to the manuscript were highlighted in green.

L22: It should be "contributions" instead of "contribution"

Thanks for pointing this out. It is now corrected.

L52: It should be "air pollutant emissions" instead of "air pollutants emissions". There are many other English grammar errors in the text part. Please fix them when you revise your manuscript.

Thanks for pointing this out. It is now corrected. In addition, we have checked the full text and corrected the grammar errors. Please check those changes as highlighted in the revised submission.

Lines 46-56: About the background and study of air pollution during the Hangzhou G20 summit, some results have been published. Regarding impact of effectiveness of short-term emission control schemes on air quality in Hangzhou during the 2016 G20 Summit, the paper of Li et al ( Li P, Wang L, Guo P, et al. High reduction of ozone and particulate matter during the 2016 G-20 summit in Hangzhou by forced emission controls of industry and traffic. Environ Chem Lett, 15:709–715, DOI 10.1007/s10311-017-0642-2) has done significant work. The authors should cite this paper.

Thanks for the suggestion. We have now cited this paper in the introduction part. The added paragraph is written as "The PM$_{2.5}$ and ozone levels in Hangzhou during the G-20 Summit were considerably lower than previous by using the WRF-CMAQ modeling system (Li et al., 2016)".

Lines95-96: Before line96, it will be good to add a paragraph as background to summarize the current status of air pollution study in Hangzhou such as the paper of Yu S, Zhang Q, Yan R, et al. Origin of air pollution during a weekly heavy haze episode in Hangzhou, China [J]. Environmental Chemistry Letters, 2014, 12(4):543-550.

Thanks for the suggestion. The background of the status of air pollution study in Hangzhou was added as "Air pollution has become serious in the Yangtze River Delta region because of the fast development of urbanization and industrialization over the past few years. Hangzhou is one of the major cities in the Yangtze River Delta, a large number of economic activities have made its air pollution increasingly prominent. Although some air pollution control measures have been adopted in recent years and the overall air quality in Hangzhou has been improved, air pollution incidents still occurred frequently. The average mass concentration of $PM_{2.5}$ were 118 μg m$^{-3}$ during a haze period and the contribution of SIA species to $PM_{2.5}$ mass increased to almost 50% under haze and fog conditions in Hangzhou (Jansen et al., 2014). Iron/steel manufacturing and secondary aerosol were the main sources for fine particles (Liu et al., 2015a). In addition to the local emissions, it is also found that air mass pathways and cross-border transports controlled high $PM_{2.5}$ concentrations and formation in Hangzhou (Yu et al., 2014). The $PM_{2.5}$ and ozone levels in Hangzhou during the G20 Summit were considerably lower than previous by using the WRF-CMAQ modeling system (Li et al., 2017b).

**Response to Reviewer Comments #3**

This manuscript presents analyses of aerosol and meteorological observations taken at an urban site in Hangzhou, China, before, during, and after the G20 Summer in Sep 2016. The measurements have rich datasets of aerosol speciation, which allows for the use of species correlation to diagnose transport and source characteristics. In combination with back trajectory analysis, the authors attempted to estimate the contribution of local, regional, and continental scale transport to the observed variations in aerosol concentrations at that site. The paper is overall well-written with solid analysis. I have two major concerns about the scale the measurements would represent and robustness of trajectory analysis.

We thank the referee for his or her insightful and helpful comments and suggestions to improve the quality of this manuscript. Please find below our point-by-point reply to the specific comments. All the changes made to the manuscript were highlighted in green.

First, the paper did not specify the resolution of the meteorological data that drive the HYSPLIT back trajectories. On line 157-158, it says the GDAS is used. The GDAS data has a horizontal resolution of 1 degree by 1 degree, roughly 100 km x 100 km. Thus, it is possible for the GDAS-driven back trajectories to distinguish between local and regional transport. The study domain shown in Figure 1 (right), which is on the scale of 300 km at the maximum, will be covered by only a few grid boxes. This shortcoming would make all the related discussion of transport patterns and source location attributions highly uncertain. I think the authors should use a much finer meteorological dataset to drive the back trajectory model, e.g. from WRF with horizontal resolutions of a few km at least.

Thanks for the comment. The GDAS reanalysis dataset used in our original backward trajectory analysis was indeed 1 degree by 1 degree. We totally agree with the reviewer that the resolution of the input meteorological data is important for deriving the transport patterns and a finer-resolution back trajectory analysis will be more accurate, e.g. WRF simulation with fine horizontal resolutions. However, we are not the expertise in the WRF model, the simulation of meteorology seems to be beyond the scope of this study. We hope that the reviewer could understand our difficulties in this issue. In addition, although the Yangtze River Delta region in Figure 1 is on the scale of 300 km, the backward trajectories reached much farther areas as shown in the original Fig. 6 & Fig. 8. After we consulted with the modeling expertise, the modeling domain should cover at least 20 degrees by 20 degrees. If WRF is conducted at a high resolution, e.g. 9km, the grid cell should be at least 200 by 200, which are highly computational

In this regard, we have redrawn all the air mass trajectories by using the GDAS reanalysis dataset with the resolution of 0.5 degree by 0.5 degree. Although this resolution is still coarse, we think this option should be valid based on the vast areas of Eastern China. The replotted figures are shown in the revised submission. As we compare the original Fig. 6 & Fig. 8 with the new plots by using the finer resolution GDAS dataset, there are some slight differences. However, the patterns of trajectory cluster analysis and WCWT are consistent between the two simulations with different resolutions. Thus, our major conclusions on the regional and long-range transport are not affected.

In the revised manuscript, we have indicated clearly about the information of meteorology dataset in the methodology section. The related paragraphs are also rewritten based on the new trajectory simulation results.

Second, I found the sea breeze related discussion groundless. First, they do not provide concrete evidence there was indeed sea breeze circulation during the study period. Sea breeze is a localized circulation pattern on the scale of a few kilometers with distinctive diurnal reversals of winds: offshore flow during the day and onshore during the night. The authors did not establish any of these patterns in the paper. I actually doubt there exists a sea breeze circulation in Hangzhou. Second, given the sea breeze as a local circulation pattern with wind reversals, it is unlikely one can detect continental outflow from the sea breeze without careful analysis of the flow patterns or modeling. What happen more often with sea breeze is the recirculation of local pollutions, as often seen in coastal cities around the globe. Offshore winds at night and during early morning push urban pollutions to the shallow marine boundary layer, which then recirculate back to the urban area in the afternoon by the offshore flow. So, if the authors could establish there was indeed sea breeze circulation during the study period, the follow-up discussion should be restricted to the local scale circulation, rather than the continental background. The local scale focus also reinforce my first comment that a finer-resolution back trajectory analysis should be used.

Thanks for pointing out this sea breeze issue and explain very clearly about the phenomenon of sea breeze and its impact on the air quality of coastal cities. We admit that our understanding of the sea-land breeze was indeed insufficient and thus may improperly explain the movement of air masses in the S4 stage. As mentioned by the reviewer, sea-land breeze is a medium-scale circulation caused by the difference in sea-land heat near the coast and it is a localized circulation pattern on the scale of only a few kilometers.

We have checked the movement of air masses according to the 48-hr backward trajectories as shown in Fig. 7b. It could be seen clearly that the air masses travelled from Hebei and Shandong provinces and finally reached Hangzhou by passing over the ocean. The movement of the air masses was on the scale of several hundred kilometers, thus precluding its nature of sea breeze.

In this regard, we agree with the reviewer that the identification of sea breeze was incorrect in this study. In the revised manuscript, we have changed the term "sea breeze" as "oceanic air masses" to avoid any confusions. We thank again for the referee's careful reviews.

Overall, we would like to thank for all three referees' insightful comments and suggestions to greatly improve the quality of this manuscript.

[revised manuscript text omitted]